# Fermentation Patterns, Methane Production and Microbial Population under In Vitro Conditions from Two Unconventional Feed Resources Incorporated in Ruminant Diets

**DOI:** 10.3390/ani13182940

**Published:** 2023-09-16

**Authors:** Karina A. Araiza Ponce, J. Natividad Gurrola Reyes, Sandra C. Martínez Estrada, José M. Salas Pacheco, Javier Palacios Torres, Manuel Murillo Ortiz

**Affiliations:** 1Faculty of Veterinary Medicine and Animal Science, Juarez University of the State of Durango, Durango 34126, Mexico; karii_araiza@hotmail.com (K.A.A.P.); javipaltor30@gmail.com (J.P.T.); 2Interdisciplinary Research Center for Integral Regional Development, National Polytechnic Institute, Durango Unit, Durango 34126, Mexico; natigre1@hotmail.com (J.N.G.R.); con_sandra@hotmail.com (S.C.M.E.); 3Scientific Research Institute, Juarez University of the State of Durango, Durango 34126, Mexico; jsalas_pacheco@hotmail.com

**Keywords:** methane, ammonia, *Leucaena leucocephala*, pricky pear, microbial population, in vitro fermentation

## Abstract

**Simple Summary:**

The production of greenhouse gases (GHG) from livestock and its impact on climate change are a major concern worldwide. It has been reported that enteric methane is the most important GHG emitted in ruminant production systems at a farm scale, accounting for approximately 50 to 60%. Many attempts have been made to modify fermentation ruminal and decrease methane production. It has recently been shown some plants, leaves, fruits, and roots reduce methane production in the rumen. Therefore, this study was conducted to investigate the inclusion of *Leucaena leucocephala leaves* (LLL) and prickly pear (PP) (*Opuntia ficus-indica*) in ruminant diets, on ruminal fermentation patterns, methane production, and microbial population under in vitro conditions. The results showed that the inclusion of *Leucaena leucocephala* in the diet decreased the concentrations of methane (CH4) and also decreased the constant rate of degradation of dry matter, ammoniacal nitrogen (NH3-N), and microbial biomass synthesis (MBS). In addition, *Opuntia ficus-indica* recorded higher potential gas production and higher dry matter intake compared to *Leucaena leucocephala*.

**Abstract:**

In this study, four experimental treatments were evaluated: (T1) alfalfa hay + concentrate, (50:50%, DM); (T2) alfalfa hay + *Leucaena leucocephala* + concentrate, (30:20:50%, DM); (T3) alfalfa hay + prickly pear + concentrate, (30:20:50%, DM); and (T4) alfalfa hay + *Leucaena leucocephala* + prickly pear + concentrate, (30:10:10:50%, DM). NH3-N concentrations in T2 and T4 decreased when replaced with alfalfa hay in 20 and 10%, respectively. Treatments did not affect the concentration of total volatile fatty acids (TVFA) between T3 and T4 (*p* > 0.05), while the concentrations among T1 and T2 were different (*p* < 0.05). T2 showed a reduction of 25.5% in the methane production when compared to T1 (*p* < 0.05). The lowest concentrations of protozoa were observed in T2 and T4, which contained *Leucaena leucocephala* (T2) and *Leucaena leucocephala* + prickly pear (T4) (*p* < 0.05). The highest concentration of total methanogens was recorded in T1 and was different in T2, T3, and T4 (*p* < 0.05). *Leucaena leucocephala*, at an inclusion percentage of 20%, decreased the methane when compared to T1, whereas prickly pear increased methane production in relation to T1.

## 1. Introduction

The growing world population demands food that includes proteins of animal origin, such as beef and milk. The intensive production of meat and milk worldwide requires the supply of fodder and energy concentrates to livestock. However, ruminal fermentation of forages and concentrates produces high amounts of methane (CH4). Therefore, considerable efforts have been devoted to finding alternative sources of forage that contribute to satisfying the nutritional requirements of ruminants and that contribute to the reduction in methane production in the rumen [1]. The CH4 is produced from carbon dioxide and hydrogen as a by-product in rumen fermentation. Agriculture accounts for about 47% to 56% of total anthropogenic methane emissions [2]. It is known that from the agricultural sector, dairy, bovine, caprine, and ovine livestock contribute substantially to the increase in CH4 production using the continuous process of ruminal fermentation. Hence, livestock activity contributes to the emission of greenhouse gases (GHG), contributing almost 30% of the total anthropogenic methane emissions into the atmosphere [3]. It has recently been recognized that due to their secondary metabolite content, some plants, leaves, fruits, and roots reduce methane production in the rumen [4]. In vitro studies have shown that secondary metabolites such as tannins have antimethanogenic activity, either directly by inhibiting methanogens or indirectly by attacking protozoa [5]. Plants with bioactive compounds (saponins and tannins) to modify fermentation and rumen inhibition of methanogenesis are one option and are generally safe, cheap, and readily available [6]. In this way, some plants, such as LLL and PP, can be used as alternative sources of forage in ruminant feeding. Moreover, the LLL is a highly available legume species commonly used as forage for ruminant feeding in North and South America, Asia, Australia, Africa, and the Pacific Ocean Island [7]. Additionally, PP has recently been introduced into diets and supplements to maintain ruminant body condition during dry periods [8]. Although numerous studies on the effects of plants with a high tannin content, such as LLL, on rumen fermentation have found reductions in enteric methane production, the mechanisms of how the tannin content of these plants reduces methane production in vitro are still unclear. It is hypothesized that the incorporation of LLL and PP into ruminant diets will decrease methane production in vitro and alter the rumen microbial population. Therefore, the objective of this study was to evaluate the effects of incorporating LLL and PP in ruminant diets on gas and methane production, rumen fermentation patterns, and the amount of methanogens during in vitro fermentation.

## 2. Materials and Methods

### 2.1. Location of Area Study and Ethical Procedure

The experiment was carried out in the animal metabolic unit and nutrition laboratory of the Faculty of Veterinary Medicine and Animal Science of the Juárez University in Durango (Mexico). Surgical procedures and management of rumen fistulated steers that were used to obtain rumen fluid were performed in accordance with the guidelines established by the Animal Protection Committee of the State of Durango (Mexico) and in accordance with the Official Mexican Standard NOM-062-ZOO-2019 [9].

### 2.2. Plant Collection, Sampling, and Chemical Analysis of Leucaena leucocephala Leaves and Prickly Pear

Samples of LLL and PP were collected in a silvopastoral pasture and in medium-sized arbosufrutescent rangeland, respectively. In general, the climate of this region is semi-arid, where the harsh conditions of drought are caused by the lack of rain. The lowest temperature is observed in winter, around 5 °C; while in summer, it fluctuates between 25–35 °C. The average annual rainfall is about 350 mm, distributed over a period of 60 rainy days during the summer. The LLL and PP leaf samples were dried at 40 °C in a forced-air oven for 72 h and ground via a 1 mm mesh prior to chemical analysis and in vitro assays. The chemical composition of forage resources is shown in Table 1.

### 2.3. Chemical Composition of the Forage Sources and Experimental Treatments

In each experimental diet, alfalfa hay, LLL, and PP were used as a forage source. The chemical composition of the forage sources is shown in Table 1.

Four treatments were evaluated: (T1) alfalfa hay + concentrate, (50:50%, DM); (T2) alfalfa hay + LLL + concentrate, (30:20:50%, DM); (T3) alfalfa hay + PP + concentrate, (30:20:50%, DM); and (T4) alfalfa hay + LLL + PP + concentrate (30:10:10:50%, DM).

### 2.4. Nutritional Composition of the Experimental Treatments

The nutritional composition of the experimental treatments is shown in Table 2. Samples from each experimental treatment were analyzed in triplicate for dry matter (DM), organic matter (OM), crude protein (CP), and ether extract (EE) [10].

Analysis of neutral detergent fiber (NDF), acid detergent fiber (ADF), and acid detergent lignin (ADL) were determined using the filter bag technique with a fiber analyzer (ANKOM Technology, Fairport, NY, EE. UU.). The total carbohydrate contents (TCH) were calculated according to the equation proposed by Sniffen et al. [12]: %TCH = 100 − %CP + %EE + %ash), while the content of non-fibrous carbohydrates (NFC) was calculated from the difference between %TCH and %NDF. The true in vitro digestibility of dry matter (TIVDMD) was determined using a Daisy incubator (ANKOM Technology, Fairport, NY, EE.UU.). Total tannins were calculated as the difference between total phenols and non-tannin phenols. The condensed tannins were measured by the HCL-butanol method [13].

### 2.5. In Vitro Gas and Methane Production Parameters

In vitro gas production was measured using the ANKOM gas production system. Rumen fluid was collected approximately 3 h after morning feeding from two steers with rumen fistula. Rumen fluid was immediately squeezed through four layers of gauze and transported to the laboratory in a sealed thermos. The resulting ruminal fluid was purged with deoxygenated CO_2_ before using it as inoculum. Approximately 1 g of dried and ground samples from each treatment were weighed and placed in glass modules. Rumen fluid buffered with McDougall’s buffer (20 mL) was pipetted into each module. Gas production was recorded after 2, 4, 6, 8, 12, 16, 24, 36, 48, 72, and 96 h of incubation. The accumulated production of gas (Y) in time (t) was adjusted to the model proposed by McDonald, [14]: GP = a + b × (1 − exp − Kd × (t − L); where GP = gas production, a = Gas production from the rapidly soluble fraction, b = Gas production from the slowly degradable fraction, (a + b) = potential gas production, Kd = Constant rate of gas production, t = incubation time, L = delay time. Gas relative production (GRP) was estimated with the following model: GRP (mL/g MS) = a + (bc/(Kd + Kp)) e-KpL [15], where: a, b, Kd and t were previously defined. The passage rate (Kp) was estimated from the model proposed by Haugen et al. [16]: Kp (%/h) = 0.07 IVDMD (%) − 0.20. To measure CH_4_ and CO_2_ production, once the incubation period was over at 24 h, The pressure release valve of each glass module was opened for 2 s, and the gas released in each module was passed through a tube and connected to a portable gas analyzer to measure CH_4_ according to procedures proposed by the equipment manufacturer (GEMTM5000, LANDTEC, Dexter, MI, USA).

### 2.6. Dry Matter Intake and In Vitro Degradability Parameters of Dry Matter and Neutral Detergent Fiber

Dry matter intake (DMI) was estimated according to Khazaal et al. [17] using the following model: DMI (g DM/kg LW75) = 18.9 + 0.23 (a + b) + 687 (Kd) + 0.11 (CP); while the kinetics of DM and NDF degradation were performed in the Daisy II incubator (ANKOM, Technology Corp., Fairport, NY, USA, EE. UU.). Bags with 2.0 g of each treatment (3 replicates) were incubated in a module (21 modules/flask) containing ruminal fluid combined with a buffer solution (1:4, vol/vol). The ruminal fluid was obtained from two steers cannulated in the rumen, which were fed with a diet containing 60% oat hay and 40% concentrate. Degradation patterns were recorded after 0, 4, 8, 12, 24, 36, 48, 72, and 96 h of incubation. The bags were removed from each module at the defined incubation times, then washed with cold water and processed in the ANKOM fiber analyzer (Fibertec 2010 (Tecator Comp., Macedon, NY, USA)) for the determination of dry matter and neutral detergent fiber (NDF). The degradation curves of DM and NDF at the different incubation times were adjusted to the following model proposed by McDonald [14]: Dt = a + b × (1 − exp(−Kd × (t − L); where Dt = degradability; a = rapidly soluble fraction; b = slowly degradable fraction, Kd = degradation rate constant, t = incubation time, L = latency time. The effective degradability (ED) was estimated as ED = a + b × (Kd/(Kd + Kp) [18]; whereas mean retention time in the rumen (MRTR) from the following model: MRTR (h) = [(1/kp) + 10] × 0.75 [16].

### 2.7. True Digestibility In Vitro of Dry Matter, Rumen Fermentation Patterns and Synthesis of Microbial Biomass of the Experimental Treatments

True digestibility in vitro of dry matter (TDIVDM) was determined using a Daisy incubator (ANKOM Technology, Fairport, NY, USA, EE. UU.). Bags with 2.0 g of each experimental treatment (3 replicates) were incubated in a module (3 bags/module) containing ruminal fluid combined with a buffer solution (1:4, vol/vol). Rumen fluid was obtained from two steers cannulated in the rumen fed a diet containing 50% alfalfa hay and 50% concentrate. The DM digestibility was recorded after 48 h of incubation. Percent weight loss was determined and recorded as the TDIVDM. After 24 h of incubation, two samples (5 mL) were taken of the liquid of the glass modules. The first subsample was acidified with 0.3 mL of 50% H_2_SO_4_, and the second subsample with 2.5 mL of 25% metaphosphoric acid. Both subsamples were immediately frozen at −40 °C and then analyzed for ammonia nitrogen (NH3-N) and total volatile fatty acids (TVFA), respectively [19]. The microbial biomass synthesis yield (MBS) and partition factor (PF) were calculated using the TDIVDM (mg) and the volume of gas registered at 24 h (GP24) as follows manner: MBS (mg^−1^ g DM) = TDIVDM (GP24 × 2.25); PF = TDIVDM/GP24 [20].

### 2.8. Rumen Microbial Population

For the extraction of DNA from each experimental treatment, rumen fluid was obtained from two steers with rumen fistulation, fed with alfalfa hay and concentrate in a 50:50 ratio, which were fed twice a day at 08:00 and 16:00 h. Rumen fluid was collected 4 h after morning feeding in a thermos and transported directly to the laboratory. Rumen fluid was filtered through four layers of cheesecloth and kept at 39 °C in a CO_2_ atmosphere. In glass modules of the ANKOM gas production system, 1 g of the ground samples of each treatment were introduced. Immediately, 125 mL of ruminal fluid and a buffer solution prepared according to Menke and Steingass were added [11]. All the glass modules containing the incubation medium and the treatment samples were incubated at 39 °C for 24 h. After the 24 h incubation was complete, 50 mL of liquid was collected from each glass module and placed in tubes to centrifuge at 20,000× *g* for 30 min. The supernatant was discarded, and 0.5 g of the residue was immediately taken for deoxyribonucleic acid extraction.

#### 2.8.1. Extraction of Rumen Microbial DNA

Deoxyribonucleic acid extraction was performed using the method described by Rojas et al. [21]. DNA concentration was calculated using a NanoDrop 2000 (Thermo Scientific, Waltham, MA, USA, EE.UU.), and DNA integrity was confirmed by agarose gel electrophoresis. The DNA samples obtained were stored at −80 °C until the quantitative analysis of microbial DNA. The YATP (g mole microbial cells −1 ATP) was calculated according to Czerkawski [22].

#### 2.8.2. Quantitative Analysis of Real-Time PCR Populations

Microbial DNA was amplified from total DNA with specific primers for each population. The sequence of the primers used for the detection of bacteria and total methanogens is shown in Table 3. The primers used for the detection of total bacteria and methanogens were *16S rRNA* and *mcrA*, respectively [23]. The specificity of the primers was verified with the conventional PCR technique using the Multigene Labnet 96-well thermal cycler (Labnet Corporation, Inc., Global, Alcobendas, Madrid, España). The number of copies was calculated from the formula proposed by Marconell [24], while the absolute quantification was obtained with the equation proposed by Angarita et al. [25].

### 2.9. Protozoa and Cellulolytic Bacteria

The estimation of the population of protozoa in the ruminal fluid was carried out by diluting 8 mL of ruminal fluid with 16 mL of formal saline solution (one part of 37% formalin and nine parts of 0.9% saline solution) and counting the protozoa under an optical microscope (10×) using a Neubauer camera [26]. To assess the bacterial population, ruminal fluid samples were diluted 1:3 in formal saline and again diluted 103 in formal saline. Crystal violet (20 mL) was added to 200 mL of this solution, and the stained bacteria were read under light microscopy (40×) using a Neubauer chamber [27].

### 2.10. Statistical Analysis

All data were submitted to a completely randomized design, and the significance of the differences between means was determined using Tukey’s multiple range test. Differences at *p* < 0.05 were considered statistically significant. All analyzes were performed using SAS [28], from the following statistical model:Yij = µ + ti + eij
where Yij is the response variable, µ is the overall mean, ti is the treatment effect, and eij is the error due to the j-th replicate of the i-th normally distributed treatment with zero mean and constant variance.

## 3. Results

### 3.1. In Vitro Gas and Methane Production

Fractions “a” and “b” were different between treatments (*p* < 0.05), being higher for the control treatment (T1) in relation to T2, T3, and T4 (Table 4). The Kd value was affected by the treatments (*p* < 0.05).

Lower Kd value for T2 (4.0 mL^−1^ h), T3 (6.0 mL^−1^ h), and T4 (4.5 mL^−1^ h) was observed, and the highest Kd was obtained with the control treatment (8.0 mL^−1^ h) (*p* < 0.05). Despite the nutrients supply by LLL leaves and PP in T2, T3, and P4, the potential gas production gas (PGP) and relative gas production (RGP) values were lower than the control treatment where alfalfa hay was a main source of forage (*p* < 0.05). Differences were observed among treatments in the methane (CH_4_) and carbon dioxide CO_2_) productions (*p* < 0.05). Treatment with LLL (T2) showed a reduction of 25.5% in methane production (*p* < 0.05) when compared to control treatment (T1), whereas carbon dioxide production showed an increase of 28.6% with T2 when compared to control treatment (T1) (*p* < 0.05). Moreover, there were differences between T1 and T2 in the CO_2_:CH_4_ ratio (*p* < 0.05). The CO_2_:CH_4_ ratio showed an increase of 74.0% with T2 when compared to the control treatment (T1) (*p* < 0.05).

### 3.2. Dry Matter Intake, In Vitro Degradability Parameters of Dry Matter and Neutral Detergent Fiber

Dry matter intake (DMI) was higher for T1 compared to the other treatments (*p* < 0.05) (Table 5). Differences were observed between treatments in the values of rapidly degradable fraction of dry matter (aDM), slowly degradable fraction of dry matter (bDM), potential degradability of the dry matter (PDDM), effective degradability of dry matter (EDDM) and degradation rate constant of dry matter (kdDM) (*p* < 0.05), except to rate passage of dry matter (kpDM) in T2, T3, and T4 (*p* > 0.05). Treatment control (T1) showed an increase of 37.0% in PDDM and 68.7% in EDDM (*p* < 0.05), respectively. Differences were observed between treatments in the values of rapidly degradable fraction of neutral detergent fiber (aNDF) (*p* < 0.05). Furthermore, the slowly degradable fraction of neutral detergent fiber (bNDF), potential degradability of neutral detergent fiber (PDNDF), and effective degradability of neutral detergent fiber (EDNDF) values were affected by treatments (*p* < 0.05). The highest value of the degradation rate constant of neutral detergent fiber (KdNDF) was recorded in T1 and the lowest in T2 (*p* < 0.05). LLL and PP did not induce any effect on the rate passage of neutral detergent fiber (KpNDF) (*p* > 0.05). Additionally, the highest value of KpNDF was recorded in T1 and the lowest in T2 and T4 (*p* < 0.05). The longer values of the mean rumen retention time of dry matter (MRRTDM) and mean rumen retention time of neutral detergent fiber (MRRTNDF) were observed in T2, and the slowest values in T1 (*p* < 0.05).

### 3.3. True Digestibility In Vitro of Dry Matter (TDIVDM)), Ruminal Fermentation Patterns and Microbial Biomass Synthesis

There were differences among treatments in the TDIVDM values (*p* < 0.05) (Table 6). The highest values of TDIVDM were recorded in T1 and the lowest in T2 (*p* < 0.05). At the same time, the concentrations of NH3-N were significantly affected by the treatments (*p* < 0.05). In our study, the concentrations of NH3-N in T2 and T4 decreased when LLL replaced alfalfa hay by 20 and 10%, respectively. Moreover, the concentration of total volatile fatty acids (TVFA) in the rumen liquor was statistically similar between T3 and T4 (*p* > 0.05), while the concentrations between T1 and T2 were different (*p* < 0.05). Acetate and propionate ruminal concentrations were affected by the treatments (*p* < 0.05). Acetate concentrations decreased when LLL replaced alfalfa hay in 20% (T2) and 10% (T4). However, propionate concentrations increased with both treatments. The highest values of microbial biomass synthesis (MBS) were recorded in T1 and the lowest in T2 (*p* < 0.05). Control treatment (T1) showed an increase of 23.0%, 12.0%, and 14.0% in relation to T2, T3, and T4, respectively (*p* < 0.05). Partition factor (PF) values were statistically similar between T2, T3, and T4 (*p* > 0.05) but different from T1 (*p* < 0.05).

### 3.4. Rumen Microbial Population after In Vitro Incubation with Rumen Fluid

No differences were observed between treatments in the total number of bacteria and cellulolytic bacteria (*p* > 0.05) (Table 7). However, there were differences between treatments in the population of protozoa (*p* < 0.05). The lowest concentrations of protozoa were observed in T2 and T4, which contained LLL (T2) and LLL + PP (T4) (*p* < 0.05). The highest concentration of total methanogens was recorded in T1 and was different in T2, T3, and T4 (*p* < 0.05). YATP values were not affected by treatments (*p* > 0.05).

## 4. Discussion

### 4.1. In Vitro Gas and Methane Production

Low in vitro gas production parameters observed in T2 and T4 could be partly explained by the negative effect of tannins on ruminal fermentation [29]. The values of the in vitro gas production parameters obtained in the current study are in partial agreement with the findings of Khazaal et al. [30] and Torres et al. [31], who evaluated in vitro conditions phenolic compounds and alfalfa hay in addition to concentrates in a 50:50 range, respectively. Despite the nutrient supply of LLL leaves and PP in T2, T3, and T4, the “a”, “b”, “GPP”, and “Kd” values were lower than the control treatment where alfalfa hay was a main source of forage, which could be due to the fact that rumen microbes were stimulated, as well as the digestibility of the incubated substrate, resulting in better gas production kinetics [32]. As regards to the decrease in CH4 production recorded in T2 and T4 could be attributed to the content of condensed tannins (CT) Beauchemin et al. [33]. This suggests that CT is at least partially responsible for this effect. According to Soltan et al. [34], LLL inhibits CH4 production both in vitro and in vivo conditions. The reduction in CH_4_ production is attributable not only to CT but could be partly due to differences in other components of the diets, mainly cell wall components [35]. There are two mechanisms to reduce enteric CH_4_ production in ruminants through CT supplementation: (a) indirectly through reduced fiber digestion, decreasing H and methane production through the CO_2_ pathway, and (b) directly by inhibiting the growth of methanogens [36]. The highest structural carbohydrate contents were recorded at T2 and T4; the better CO_2_ production efficiency registered in T2 and T4 compared to the other treatments could be explained by the degradation of the structural carbohydrates of both diets. Cellulolytic bacteria that hydrolyze the structural carbohydrates of the cell wall generate acetate and CO_2_ as final products [37]. Greater production of acetate by ruminal fermentation causes a greater availability of CO_2_ [38].

### 4.2. Dry Matter Intake, In Vitro Degradability Parameters of Dry Matter and Neutral Detergent Fiber

Differences observed between treatments in DMI could be attributed to the NDF contents of the experimental diets [34]. Our results do not agree with Paengkoum [39], who found higher values of DMI when supplementing diets based on corn silage with LLL. The highest values of aDM, bDM, DPDM EDDM, and KdDM recorded in T1 may suggest the availability of nutrients provided by carbohydrates and proteins from alfalfa hay [40]. Therefore, the higher values obtained for the aDM, bDM, and EDDM fractions will indicate a better nutrient availability for rumen microorganisms. Furthermore, the slower value of KdMS recorded in T2 indicates that the amount of energy that can be extracted from the diets during the time it remains in the rumen is low. Our results are in partial agreement with the DM degradability parameters reported by Mohammadabadi et al. [41], who investigated the effect of replacing alfalfa hay with L. leucocephala leaves in proportions of 25, 50, and 100%. EDDM values registered in T1 could be associated with the high contents of NDF and TC, which become severe limitations to improve the ruminal digestion of the nutrients contained in the treatments and negatively affect the dry matter degradability. MRRTDM’s higher values and lower KpDM values observed in T2 and T4 can also be explained by the NDF contents. Minson [42] has pointed out that when the content of the cell walls increases, the KpDM decreases and the MRRTDM increases proportionally, causing the cellulose and hemicellulose to ferment slowly, and this causes the physical filling of the rumen [43]. The highest kdNDF value was recorded in T1 and the lowest in T2 (*p* < 0.05). This suggests that T2 was being broken down at a slower rate. In the present study, aNDF, bNDF, EDNDF, and kdNDF decreased when LLL replaced alfalfa hay by 20%. This may be due to the presence of anti-nutritional factors such as saponins and tannins in LLL [44]. KpNDF value was higher in the treatment with alfalfa hay as a forage source (T1) (*p* < 0.05), while the other three were similar (*p* > 0.05). The decrease in the in vitro degradability parameters of NDF observed in T2 (aNDF, bNDF, EDNDF, and kdNDF) does not agree with the results obtained by Barros et al. [45], who found higher values, including LLL at 20% of the diet. These differences between both studies can be attributed to differences in the methods to determine degradability [46].

### 4.3. True Digestibility In Vitro of Dry Matter, Ruminal Fermentation Patterns, and Microbial Biomass Synthesis

TDIVDM value in T1 could be attributed to a high synthesis of microbial biomass [47], while in T2, it could be explained by the lower contribution of ammonia and non-fibrous carbohydrates (NFC) for microbial growth [48]. The decrease in ammonia concentrations in T2 and T4 could be explained by the tannin content of the diets. There is general agreement that tannins decrease the degradation of proteins provided by the diets, mainly via the formation of tannin-protein complexes, which help to decrease the concentration of NH3-N [49]. Despite this trend, the NH3-N values observed in this study are within the optimal range to maximize microbial growth in the rumen, which is reported between 5 and 10 mg/dL [50]. Ruminal NH3-N concentrations are consistent with the results found by Kang et al. [51] when evaluating LLL in ruminant diets. The decreased acetate in T2 and T4 and increases in propionate concentrations in both treatments can be attributed to the contents of structural carbohydrates (NDF) as well as non-fibrous carbohydrates (NFC) supplied by the experimental diets [52]. In fact, previous studies have consistently reported a decrease in the molar ratio of acetate and increases in the ratio of propionate under in vitro conditions using high-starch concentrates [53] and high-fiber forages [54]. Generally, the results of the in vitro fermentation patterns obtained in this study are consistent with the findings of previous studies in which alfalfa hay, LLL, and PP were evaluated as sources of forage in ruminant diets [55]. MBS values recorded in T1 could be attributed to a greater supply of NH3-N by the experimental diet. TDIVDM estimates, and MBS obtained in this study do not agree with those found by Albores et al. [56], who found higher values in TDIVMD and MBS when including various levels of LLL in ruminant diets. PF is regularly used as an indicator of substrate degradation rate, as well as in vitro gas production efficiency and microbial biomass. In this study, PF values are consistent with what was reported by Abdallah et al. [57] and were higher than the theoretically possible maximum value of 4.41 mg TDIVMD/mL of gas [58]. The increase in PF could indicate a lower partition of nutrients for the synthesis of microbial proteins [59].

### 4.4. Microbial Population after In Vitro Incubation with Rumen Fluid

Regarding the number of total bacteria, the results obtained agree with Pilajun and Wanapat [60], who reported that supplementation with tannin-rich plants did not change the total number of bacteria. However, other studies have shown that plants rich in tannins reduce the number of bacteria [61,62]. Additionally, these results agree with Longo et al. [63], who found that the diversity indices of the methanogenic community did not change when LLL or other tannin-rich plants were supplied. Pineiro et al. [64], when evaluating the LLL in heifers fed with low-quality forage, did not observe changes in the concentration of protozoa. However, Barros et al. [44] reported that rumen protozoa decreased when ewes were fed 20% and 40% LLL. YATP values recorded in this study are within the established ranges for different diets supplied to ruminants. For a mixed-species microbial population, the estimated YATP (grams dry weight of cells formed/mole ATP spent) is 29 to 30 for growth on rich media containing preformed monomers and from 20 to 29 for growth in simple media containing carbohydrates and inorganic salts [65].

## 5. Conclusions

The results revealed that the addition of alfalfa hay to diets as a forage source (control treatment) produced the best results in degradability parameters, ruminal fermentation patterns, and microbial mass synthesis. Although the addition of LLL (T2) in the diet decreased methane production. However, the values in the in vitro gas production parameters, dry matter degradability, neutral detergent fiber degradability, ruminal fermentation patterns, and protozoa population were higher in T3 (PP) than when compared with T2 (LLL) and T4 (LLL + PP). The results obtained in this research also indicate that both unconventional forage sources can be used in ruminant diets.

## Figures and Tables

**Table 1 animals-13-02940-t001:** Chemical compositions of the three forage sources (g Kg^−1^ DM).

	Alfalfa Hay	LLL	PP
DM	897	895	900
OM	871	915	720
CP	167	213	53
NDF	450	429	483
Lignin	81	53	48
TDIVMD	557	457	515
NFC	234	670	649
TPC g tannic acid eq/kg DM	96.5	119.6	101.2
CT mg/g DM	0.40	0.98	0.51

DM = Dry matter; OM = Organic matter; CP = Crude protein; NDF = Neutral detergent fiber; TPC = Total phenolic compounds; CT = condensed tannins; TDIVMD = True digestibility in vitro of dry matter; NFC = Non-fibrous carbohydrate.

**Table 2 animals-13-02940-t002:** Nutritional composition of experimental treatments.

	**Treatments**			
**Ingredient (g kg^−1^ DM).**	**T1**	**T2**	**T3**	**T4**
Alfalfa hay	500	300	300	300
Leucaena Leaves	0	200	0	100
Prickly pear	0	0	200	100
Corn milled	350	370	280	340
Cottonseed	140	120	210	150
Minerals	10	10	10	10
**Chemical composition (g kg^−1^ DM).**				
DM	883	879	803	898
OM	904	915	874	883
CP	140	146	142	148
EE	31	27	21	24
NDF	422	473	424	461
Lignin	41	58	45	55
NFC	292	218	273	245
TPC g tannic acid eq/kg DM	105.4	122.5	95.9	106.0
CT mg/g DM	0.305	3.34	0.360	1.96
ME Mcal/kg-DM *	3.8	3.1	3.4	3.0

DM = Dry matter; OM = Organic matter; CP = Crude protein; EE = ether extract; matter; NDF = Neutral detergent fiber; ADF = Acid detergent fiber; TPC = Total phenolic compounds; CT = condensed tannins; NFC = Non-fibrous carbohydrate; * Estimated from the equation ME (Mcal kg^−1^ DM) = 2.20 + 0.136 Gas production 24 h + 0.057 CP + 0.0029 ether extract2/4.184 [11].

**Table 3 animals-13-02940-t003:** Primer sequences used to quantify total bacteria by qPCR.

**Primer Sequences Used to Quantify Total Bacteria by qPCR**		
Gene *16S rRNA*	Sequence (5′-3′)	Extension
Forward	5′CGGCAACGAGCGCAACCC3′	130 bp
Reverse	5′CCATTGTAGCACGTGTGTAGCC3′	
**Primer Sequences Used to Quantify Total Mthanogens by qPCR**		
Gene *mcrA*	Sequence (5′-3′)	Extension
Forward	5′TTCGGTGGATCDCARAGRGC3′	128 bp
Reverse	5′GBARGTCGWAWCCGTAGAATCC3′	

*mcrA* = Methyl-coenzyme M reductase; bp = base pairs.

**Table 4 animals-13-02940-t004:** In vitro gas parameters and methane production of experimental treatments.

	Treatments				SEM	*p* < Value
	T1	T2	T3	T4		
a (mL 200 mg^−1^ DM)	16.1 ^a^	7.1 ^d^	12.3 ^b^	9.4 ^c^	1.8	0.001
b (mL 200 mg^−1^ DM);	102.0 ^a^	85.7 ^d^	98.3 ^b^	92.1 ^c^	2.3	0.01
PGP (mL 200 mg^−1^ DM)	118.1 ^a^	92.8 ^c^	110.6 ^b^	101.5 ^b^	1.1	0.05
RGP (mL 200 mg^−1^ DM)	110.1 ^a^	85.8 ^d^	101.6 ^b^	93.5 ^c^	1.7	0.003
Kd (mL^−1^ h)	8.0 ^a^	4.0 ^d^	6.0 ^b^	4.5 ^c^	0.01	0.01
L (h);	2.5	3.3	3.1	3.1	0.33	0.120
CH_4_ (mL g^−1^ DM);	13.7 ^b^	10.2 ^d^	15.8 ^a^	12.8 ^c^	3.3	0.01
CO_2_ (mL g^−1^ DM).	74.8 ^d^	96.2 ^a^	87.5 ^c^	91.1 ^b^	1.2	0.001
CO_2_:CH_4_:ratio	5.4 ^c^	9.4 ^a^	5.6 ^c^	7.1 ^b^	0.12	0.001

^abcd^ Means within the same row with various superscripts are significantly different (*p* < 0.05). a = Gas production from quickly soluble fraction; b = Gas production from insoluble fraction; PGP = Potential gas production; RGP = Relative gas production; Kd = Gas production rate; L = Discrete lag time prior to gas production; CH_4_ = Methane; CO_2_ = Carbon dioxide; SEM = Standard error of mean.

**Table 5 animals-13-02940-t005:** Intake and dry matter in vitro degradability parameters and neutral detergent fiber.

	Treatments				SEM	*p* < Value
	T1	T2	T3	T4		
DMI (g^−1^ LW^0.75^)	73.0 ^a^	41.2 ^d^	62.2 ^b^	48.6 ^c^	2.9	0.001
aDM (mg g^−1^ DM)	27.4 ^a^	17.6 ^d^	22.1 ^b^	19.7 ^c^	2.1	0.002
aNDF (mg g^−1^ NDF)	12.4 ^a^	10.5 ^c^	11.7 ^b^	11.1 ^b^	1.8	0.04
bDM (mg g^−1^ DM)	58.2 ^a^	44.9 ^d^	51.3 ^b^	48.3 ^c^	2.0	0.004
bNDF (mg g^−1^ NDF)	71.1 ^a^	62.6 ^d^	69.8 ^b^	66.3 ^c^	2.3	0.01
PDDM (mg g^−1^ DM)	85.6 ^a^	62.5 ^d^	73.4 ^b^	68.0 ^c^	1.5	0.005
PDND_F_ (mg g^−1^ NDF)	83.1 ^a^	73.1 ^d^	81.5 ^b^	77.4 ^c^	2.1	0.05
EDDM (mg g^−1^ DM)	65.8 ^a^	39.0 ^d^	54.6 ^b^	47.0 ^c^	1.1	0.004
EDNDF (mg g^−1^ NDF)	45.3 ^a^	28.3 ^d^	42.0 ^b^	40.5 ^c^	1.6	0.05
KdDM (mg^−1^ h)	8.2 ^a^	3.3 ^d^	7.1 ^b^	5.2 ^c^	0.005	0.005
KdNDF (mg^−1^ h)	5.0 ^a^	2.0 ^c^	4.0 ^b^	4.0 ^b^	0.002	0.01
LDM (h)	2.0	2.3	2.1	2.0	0.98	0.05
LNDF (h)	3.3	4.0	3.5	3.8	1.7	0.18
KpDM (mg^−1^ h)	4.2 ^a^	3.6 ^b^	4.1 ^a^	4.0 ^a^	0.002	0.05
KpNDF (mg^−1^ h)	5.8 ^a^	5.0 ^b^	5.2 ^b^	5.0 ^b^	0.007	0.05
MRRTDM (h)	23.8 ^c^	27.7 ^a^	24.3 ^c^	25.0 ^b^	1.1	0.002
MRRTNDF (h)	12.5 ^c^	18.2 ^a^	14.6 ^b^	14.9 ^b^	2.5	0.03

^abcd^ Means within the same row with various superscripts are significantly different (*p* < 0.05). SEM = Standard error of mean.

**Table 6 animals-13-02940-t006:** True degradability in vitro dry matter, fermentation ruminal patterns and microbial biomass synthesis of experimental treatments.

	Treatments				SEM	*p* < Value
	T1	T2	T3	T4		
TDIVMD_48 h_, (mg^−1^ 100 mg DM)	667 ^a^	622 ^d^	654 ^b^	642 ^c^	2.4	0.030
pH	6.60	6.6	6.6	6.5	0.017	0.854
N-NH3, (mg dL^−1^)	12.6 ^a^	8.7 ^d^	11.5 ^b^	9.3 ^c^	0.152	0.024
TVFA, (mM/L)	10.6 ^a^	6.5 ^d^	7.9 ^b^	7.5 ^b^	0.281	0.002
	Volatile fatty acids (molar%)					
Acetate	66.8 ^d^	72.0 ^a^	67.7 ^c^	70.5 ^b^	0.161	0.003
Propionate	24.4 ^a^	18.2 ^d^	22.3 ^b^	19.3 ^c^	0.674	0.033
Butyrate	5.7 ^b^	9.1 ^a^	9.3 ^a^	9.0 ^a^	0.247	0.027
A:P ratio	2.7	3.9	3.0	3.6	0.143	0.911
MBS (mg^−1^ g DM)	165.2 ^a^	132.2 ^d^	147.4 ^b^	145.1 ^c^	1.13	0.007
PF (mg TDMD/mL gas)	6.0 ^b^	6.5 ^a^	6.5 ^a^	6.3 ^a^	1.05	0.050

^abcd^ Means within the same row with various superscripts are significantly different (*p* < 0.05). TDMD_48 h_ = True degradability dry matter; TVFA = Total volatile fatty acids MBS = Microbial biomass synthesis; PF = Partition factor. SEM = Standard error of mean.

**Table 7 animals-13-02940-t007:** Ruminal microbial population of the experimental treatments after in vitro incubation with rumen fluid.

	Treatments				SEM	*p* < Value
	T1	T2	T3	T4		
Total bacteria ^1^	14.7	14.9	15.2	15.4	0.030	0.22
Celulolytic bacteria ^4^	7.6	7.6	7.6	5.0	0.076	0.98
Protozoa ^3^	16.6 ^a^	7.3 ^d^	13.3 ^b^	10.2 ^c^	0.082	0.04
Total methanogens ^2^	14.2 ^a^	13.5 ^b^	13.8 ^b^	13.6 ^b^	0.066	0.05
Methanogen:bacteria ratio	0.95	0.91	0.91	0.88	0.091	0.88
Y_ATP_ (g microbial cells mol^−1^ ATP)	24.0	22.2	24.0	23.3	0.61	0.930

^abcd^ Values with different letters in the same row are statistically different (*p* < 0.05); ^1,2^ Log [ngDNAg-1 RC]; ^3^ (×10^4^ CFU^−1^ mL); ^4^ (×10^6^ CFU^−1^ mL); RC = Ruminal content; CFU = Colony forming units. SEM = Standard error of mean.

## Data Availability

The data that support the findings of this study are available from the corresponding author upon reasonable request.

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
