# Peer review of "Fermentation Patterns, Methane Production and Microbial Population under In Vitro Conditions from Two Unconventional Feed Resources Incorporated in Ruminant Diets"

_animals, 2023, doi:10.3390/ani13182940_

Round 1

Reviewer 1 Report

General comment for the paper

Overall all this is a well written paper and is very interesting research.  I hope you take it to the cow for the next step to see if it works in vivo.

When you say 60% oat hay:40% concentrate, state what the concentrate is same thing with the 50:50.  It is similar to what you show in table 2 or is it different?

 There was a lot of use of the words “likewise” and :there were difference…” try to use other phrase like “Differences were observed”  “Similarly” “ ”Also”

 Make sure there is 1 space after “.”

 Make sure there is a space before and after ()

 Make sure there are spaces between ( P < 0.05)

Lines

Comment

21

Change to deceases and remove “the”

22

Remove “Among them” start the sentence with “It has…”

23

Change Thus the Therefore

27

Starting with constant rate of…. To the end I am not sure what you are saying here is the sample decreasing, increasing staying the same please rephrase.

38

Change to “was different from T2….”

40

For prickly pear was it a negative effect or no effect please rephrase.

53-54

Please review so other papers there is work out there now that shows it may not be this high Mitloehner is a good reference to look at.

66-67

The feeds you are talking about are common for your area, but not for the rest of the world so state where they are used the most.

72

Remove “In accordance with the above” and start the sentence with “It is…”

73

“diets will decrease the methane….”

Table 7

Make sure the numbers are superscripts and put on different lines, so it is easier to read.

418

Start the sentence with “The result….”

419

State what you mean by “best results”

The paper was well written with just a few places where the writing could be better.

Reviewer 2 Report

General comments:

This trial provides interesting information on the use of alternative feed resources in ruminants diets, in substitution with conventional forage sources like alfalfa hay.

This study could be considered to be published in Animals after minor revision. It is an interesting topic, and all the knowledge and information about new and alternative feed resources is of interest for the scientific and livestock community.

I only have some general questions:

-         How variable is the composition across time of the alternative feed resources used in this experiment?

-         And what about their content in plant secondary compounds? Can you name and describe some of them?

-         It would be interesting to refer the methane production to TVFA?

·        Specific comments 

Simple Summary:

Lines 26-28: I miss some comments about the Opuntia

Abstract:

Line 40: try to write the last sentence in a different way. It seems a little confusing to me.

Introduction:

Line 76: instead of methanogenic bacteria I would suggest saying “methanogens” or “archaea”.

Material and methods:

Table 2: why was the NDF content higher in diet 2 if it was shown on table 1 that LLL had a lower NDF content? And regarding NFC content it should also be higher in T2 if we take into account what it is shown in table 1.

Lines 128-148: in this paragraph I missed the number of replicates per treatment that where incubated. And why it was decided to use only 2 steers as rumen inoculum donors? And I also missed the basal diet that was fed to the donors?

Lines 184-197: why was used a different diet in the donors in this case?

Results:

Table 4: the value of 7.8 for CO2:CH4 ratio in treatment T2 should be revised. It should be 9.43.

Table 6: please revised the superscripts. I found some mistakes in pH and TVFA

Line 291: If you check Table 6 is all the way around.

Line 292: the same that line 291

There are no non-specific comments on the use of English.
